# Cross-sectional study of the burden and determinants of non-medical and opportunity costs of accessing chronic disease care in rural Tanzania

Anna Verjans,[1,2] Brady Hooley [ID],[1,2] Kassimu Tani,[1,2,3] Grace Mhalu [ID],[3] Fabrizio Tediosi [ID] [1,2]

¹Swiss Tropical and Public Health Institute, Allschwil, Switzerland
²University of Basel, Basel, Switzerland
³Ifakara Health Institute, Dar es Salaam, Tanzania

**Correspondence to**
Dr Brady Hooley;
brady.hooley@swisstph.ch

## ABSTRACT

**Objectives** Countries in sub-Saharan Africa are seeking to improve access to healthcare through health insurance. However, patients still bear non-medical costs and opportunity costs in terms of lost work days. The burden of these costs is particularly high for people with chronic diseases (CDs) who require regular healthcare. This study quantified the non-medical and opportunity costs faced by patients with CD in Tanzania and identified factors that drive these costs.

**Methods** From November 2020 to January 2021, we conducted a cross-sectional patient survey at 35 healthcare facilities in rural Tanzania. Using the human capital approach to value the non-medical cost of seeking healthcare, we employed multilevel linear regression to analyse the impact of CDs and health insurance on non-medical costs and negative binomial regression to investigate the factors associated with opportunity costs of illness among patients with CDs.

**Results** Among 1748 patients surveyed, 534 had at least one CD, 20% of which had comorbidities. Patients with CDs incurred significantly higher non-medical costs than other patients, with an average of US$2.79 (SD: 3.36) compared with US$2.03 (SD: 2.82). In addition, they incur a monthly illness-related opportunity cost of US$10.19 (US$0–59.34). Factors associated with higher non-medical costs included multimorbidities, hypertension, health insurance and seeking care at hospitals rather than other facilities. Patients seeking hypertension care at hospitals experienced 35% higher costs compared with those visiting other facilities. Additionally, patients with comorbidities, older age, less education and those requiring medication more frequently lost workdays.

**Conclusion** Outpatient care in Tanzania imposes considerable non-medical costs, particularly for people with CDs, besides illness-related opportunity costs. Despite having health insurance, patients with CDs who seek outpatient care in hospitals face higher financial burdens than other patients. Policies to improve the availability and quality of CD care in dispensaries and health centres could reduce these costs.

## STRENGTHS AND LIMITATIONS OF THIS STUDY

⇒ This study includes data from 35 facilities—all health centres and hospitals and a sample of dispensaries—in the Same and Kilombero districts of Tanzania, and a large sample size, which strengthen the robustness of the results.

⇒ We employed exit surveys, which minimise the recall bias related to the reported time and transportation cost variables, but do not capture those who might forgo healthcare because of financial barriers.

⇒ The main limitation of this study is the lack of reliable self-reported individual income estimates, which is why we adopted the minimum daily wage instead.

⇒ However, the advantage of using the minimum wage to value patients' time is that it equally values their time, instead of assigning a higher value to the time of patients with higher salaries.

need without suffering financial hardship'.[1] Country-level policies have so far focused on reducing the direct medical out-of-pocket (OOP) cost of healthcare services and drugs, without addressing the non-medical costs of illness such as loss of income and the cost of travelling to access healthcare.[2] Using healthcare induces costs that extend beyond the medical costs covered by health insurance and involves transportation and income losses for both patients and any accompanying caregivers.[2]

This narrow focus of social health protection and service provision is particularly problematic for health systems in low-income and middle-income countries (LMICs), which are still mainly focused on the management of communicable diseases where the household economic burden associated with illness is more readily characterised by discrete, yet still potentially catastrophic disease events.[3,4] On the other hand, despite mounting evidence of the increased burden of non-communicable

## INTRODUCTION

Universal health coverage aims to ensure 'that people have access to the healthcare they

diseases (NCDs) in LMICs, care for chronic diseases (CDs) remains centralised at hospitals and many medicines for the treatment of these conditions are not readily available at primary and secondary care facilities.[5] As a result, the financial burden of seeking CD care tends to be greater than that of seeking acute care, in that patients with CD may need to use healthcare services more frequently and seek centralised care at hospitals in order to ensure that they receive the care they require.[6 7]

The issue of burdensome non-medical healthcare costs has been studied in sub-Saharan Africa (SSA), particularly in the context of HIV and tuberculosis (TB) care.[8–10] As with NCDs, the chronic nature of HIV and TB management requires patients to visit healthcare facilities and renew prescriptions on a 1–3 month basis, meaning that patients must regularly pay for transport to the clinic and spend time accessing care that could have otherwise been spent working.[9 10] The accumulation of travel costs and productivity loss associated with illness and care seeking can therefore surpass the direct cost of care, representing 55% of costs for HIV infections and 71% of costs for HIV/TB coinfections.[10] However, the introduction of a decentralised community-based HIV care programme in Tanzania substantially reduced both the indirect and direct costs of accessing HIV care, indicating that such an intervention or policy implementation could yield promising results for reducing the indirect costs of care seeking for people living with chronic conditions.[11]

While research on the non-medical and opportunity costs of seeking NCD care in SSA is more limited, past work has indicated that these costs can be substantial for some patients, and some of them even completely forgo seeking care in order to avoid the impoverishing effect of direct medical or non-medical costs of care.[6 12–15] In Mali, the indirect costs of diabetes mellitus were estimated to make up 61% of the total costs,[6] while in rural Malawi the direct costs of NCDs formed the largest cost element, with 56.8% of the total.[16] While not directly comparable, their findings demonstrate a substantial economic burden on patients.

Non-medical costs form a barrier that can therefore affect uptake and adherence to treatment, which for chronic conditions can contribute to severe long-term effects. In this regard, prevention and early treatment of NCDs can avoid the need for more expensive treatment procedures in the future, caused by more severe illness, which usually also requires more frequent follow-up. For instance, a study found higher transportation costs and longer waiting times for antiretroviral therapy (ART) compared with pre-ART patients in rural South Africa.[8] Besides early onset of treatment, absenteeism and short-term disability among patients with CD can also be reduced by improving medication adherence.[17 18] This highlights the potential economic gain of implementing alternative care models that lower the financial burden for patients with NCD by avoiding future opportunity costs.

The human capital method (HCM) and friction cost method are two of the most commonly used methods for measuring opportunity costs. The former measures opportunity costs in terms of the value of lost income, including future productivity losses, while the latter only considers these losses up until the market finds a replacement.[19] The HCM takes a patient perspective and is often preferred over the friction cost method since it is better able to account for presenteeism, although it is sometimes criticised for overestimating the costs as it assumes that the duration of absence from work fully corresponds to the market value of those lost days.[20] However, the dependence of agricultural economies on seasonality means that sick days do not necessarily translate into lost working days.[21] Moreover, to compensate for lost working time and income, households often adopt various 'distress financing' strategies, such as borrowing money or selling assets, especially in informal economies.[21] Income losses in the agricultural sector are sometimes approximated by the minimum daily wage of the country or by taking the gross national income (GNI) per capita and transforming it into a daily value,[6 15 22–24] the latter of which may inflate opportunity costs.[25 26] The former method aims to address the lack of equity of the HCM by valuing everyone's time equally.

Previous work in Tanzania showed that people with cardiovascular risk factors incur substantial non-medical and opportunity costs, yet there is little evidence regarding the actual non-medical costs of seeking NCD care in Tanzania.[27] Considering the potential burden of non-medical costs of seeking healthcare and opportunity costs of losing workdays due to illness, this study surveyed outpatients from 35 health facilities in two rural districts in order to estimate these costs in the rural Tanzanian context.

## METHODS

This study is part of the Chronic Conditions Household and Exit Survey in Tanzania (CHEST), a cross-sectional outpatient and household survey in the rural Same and Kilombero districts conducted between November 2020 and January 2021. This survey recruited 784 household members and 1748 (We calculated the sample size based on the hypertension and diabetes prevalence for rural Tanzanian adults, and population estimates for the two districts. A modified Cochran sample size calculation (power: 0.80, significance level: 5%) resulted in a minimum sample size of 202 patients per district.[28]) outpatients above the age of 18. This study used only outpatient data on the reason for their health facility visit, time and cost of transport to the facility, time spent at the clinic and what they expect they would have earned if they had not sought care that day. The sampling strategy and full details of the CHEST survey have been described in previous work.[28]

The patient health facility exit survey was administered to adult patients at all tertiary and secondary health facilities

in both districts and a matched sample of 8 dispensaries in each of the Kilombero and Same districts. As there are eight health centres in each of these districts, the dispensary sampling was matched by randomly selecting one dispensary from each ward containing a health centre. All adult patients attending the outpatient clinic on the day of the survey were eligible for recruitment.

Outpatient clinics in hospitals and health centres typically designate 1 day a week as an 'NCD day', when a medical officer is assigned to be available to provide outpatient NCD care.[28] Therefore, for each sampled hospital and health centre, the exit surveys were conducted on an NCD day at the health facility to ensure that a sufficient number of participants with NCDs would receive the questionnaire, and on a non-NCD day at the clinic to also collect a more typical sample of people seeking outpatient care.

Rather than random or systematic random sampling of participants, this study used the more operationally efficient method of selecting and recruiting participants based on the order in which they entered the consultation room. This is demonstrated to be easier to implement than random sampling approaches, and it minimises the bias in consultation length associated with sampling patients as they leave the consultation room.[29]

The outcome of interest was the total non-medical cost of seeking healthcare services, comprising both direct and indirect non-medical costs. In this case, direct non-medical cost included transportation fare while the indirect non-medical costs included the opportunity cost associated with travel and clinic time. We excluded those individuals with missing data for the travel cost or travel or clinic time from the analysis, resulting in 1638 individuals with full cost data. We estimated the indirect non-medical cost using the HCM, using the national minimum hourly wage for the agricultural sector (512.82 TZS=~US$0.22), the largest occupational group in our sample.[30] However, some salaried participants reported lost income, so we performed a subanalysis of these data, while imputing human capital-based opportunity costs for those participants who did not provide estimates for their forgone wages (see online supplemental materials). All costs were converted to 2020 US dollars (US$1=2277 TZS).[31] For survey participants who claimed to have a chronic condition, we additionally estimated the cost of lost work over the past month due to illness, based on the number of days they reported being unable to work.

We used a mixed-effect multivariate linear regression to investigate the association between non-medical costs and insurance status and CD diagnoses, while controlling for various demographic factors such as gender, age, education, occupation, accompaniment by a caregiver, level of healthcare facility and residential proximity to the facility. We included a random intercept term to account for between-district variability, and log-transformed non-medical costs in order to maintain the assumptions of linear regression. Additionally, we included an interaction

term to examine the impact of hospital care on patients with hypertension.

On 1 January 2023, the Government of Tanzania introduced a new wage order, increasing the hourly wage of agriculture workers by approximately 40%, from 512.82 TZS to 718 TZS.[30 32] We therefore performed a sensitivity analysis to investigate the impact of higher wages on indirect health expenditure by using the new minimum wage in HCM estimates of opportunity costs.

To evaluate the impact of reduced work capacity on patients with CDs, we used negative binomial regression (Poisson regression models were tested against negative binomial models using likelihood ratio tests. Given the significant overdispersion, the negative binomial model was preferred, which is in line with analysis standards for absenteeism data.) to determine the association between engagement in care and the number of days missed from work over the last 30 days. The predictor variables of interest included the prescription of medication during the last visit, medication adherence over the last week and the presence of multiple chronic conditions, with controls for age, gender, education and occupation. We used both STATA SE V.16.1 and R (V.4.1.3) for analysis.[33 34]

### Patient and public involvement statement

We first involved the public by engaging with healthcare providers prior to the design of this study, where informal conversations with providers at rural hospitals and health centres indicated that few NCD services are available at dispensaries and health centres and that patients with NCDs must frequently be referred or self-refer over substantial distances for relatively basic NCD services and prescription medicines. These providers lamented that the need to seek CD care in hospitals forms both a substantial barrier to care that prevents would-be patients from being retained in care in a timely manner, and that those who are retained in care face substantial non-medical costs.

The findings of this study were directly presented to Tanzanian stakeholders and policy-makers via workshops and presentations in Dodoma and Dar es Salaam, where we engaged with attendees from the Ministry of Health, National Health Insurance Fund, Improved Community Health Fund and President's Office for Regional Administration and Local Government.

### RESULTS

We recruited 1748 outpatients, of which 63.73% were women, 64.87% married, 52.97% completed primary school, 53.09% subsistence farmers and a mean age of 44 years (table 1). Forty per cent were health insurance beneficiaries and about 30% had a chronic condition. Our analysis focused on chronic conditions, of which the majority are NCDs, with hypertension being the most commonly reported condition (81%). Approximately 21% of patients with CDs had more than 1 chronic condition. Patients with CDs lost an average of 5 working days

**Table 1** Descriptive statistics of the total sample

| Variable | N | Mean | SD |
|---|---|---|---|
| Sex (male) | 1748 | 36.3% | 48.1% |
| Education level | | | |
| Never attended school | 1748 | 4.9% | 21.6% |
| Some primary school | 1748 | 12.2% | 32.7% |
| Completed primary school | 1748 | 53% | 49.9% |
| Some secondary school | 1748 | 3.8% | 19.1% |
| Completed secondary | 1748 | 19.5% | 39.6% |
| Completed college education | 1748 | 4.9% | 21.5% |
| Completed university education | 1748 | 1.8% | 13.2% |
| Working (last 12 months) | 1748 | 26.8% | 44.3% |
| Occupation | | | |
| Public servant | 1748 | 5.2% | 22.2% |
| Private formal sector | 1748 | 8% | 27.1% |
| Subsistence farmer | 1748 | 53.1% | 49.9% |
| Large-scale farming | 1748 | 0.4% | 6.3% |
| Self-employed/small business | 1748 | 16.6% | 37.2% |
| Self-employed/large business | 1748 | 0.3% | 5.9% |
| Taking care of home and/or children | 1748 | 4.8% | 21.4% |
| Student | 1748 | 4% | 19.6% |
| Retired | 1748 | 4.6% | 21% |
| Other | 1748 | 3% | 17% |
| Marital status | | | |
| Married | 1748 | 64.9% | 47.8% |
| Living with partner | 1748 | 2.6% | 15.8% |
| Divorced | 1748 | 0.7% | 8.6% |
| Separated | 1748 | 4.7% | 21.2% |
| Widowed | 1748 | 11.7% | 32.1% |
| Never married | 1748 | 15.4% | 36.1% |
| Age | 1748 | 44.134 | 16.861 |
| Health insurance | 1747 | 39.5% | 48.9% |
| Any social health protection | 1748 | 47.1% | 49.9% |
| Any chronic condition | 1748 | 30.5% | 46.1% |
| Type of chronic condition | | | |
| ▶ Hypertension | 534 | 80.7% | 39.5% |
| ▶ Diabetes | 534 | 23.8% | 42.6% |
| ▶ Chronic kidney disease | 534 | 1.5% | 11.4 |
| ▶ Epilepsy | 534 | 1.3% | 10.6% |
| ▶ Asthma | 534 | 2.8% | 16.5% |
| ▶ HIV | 534 | 4.3% | 20.3% |
| ▶ TB | 534 | 0.06% | 7.5% |
| ▶ Other | 534 | 7.5% | 26.3% |
| Multiple chronic conditions | 534 | 20.8% | 40.6% |
| Prevented from working | | | |
| Completely prevented | 534 | 9.6% | 29.4% |
| Never prevented | 534 | 51.1% | 50% |

**Table 1** Continued

| Variable | N | Mean | SD |
|---|---|---|---|
| Sometimes prevented | 534 | 39.3% | 48.9% |
| Days missed work (last month) | 534 | 5.15 | 9.574 |
| Last time sought CD care | 504[1] | | |
| ▶ Within the last month | | 74.8% | 43.5% |
| ▶ Within the last 6 month | | 13.3% | 34% |
| ▶ More than 6 months ago | | 11.9% | 32.4% |
| Medicines prescribed during last visit | 534 | 89.1% | 31.1% |
| Medicines taken in the last 7 days | 534 | 68.2% | 46.6% |
| Facility level | | | |
| Dispensary | 1748 | 9.5% | 29.3% |
| Health centre | 1748 | 67.1% | 47% |
| Hospital | 1748 | 23.4% | 42.3% |
| Closest facility visited | 1748 | 81.4% | 38.9% |
| Usual facility visited | 1748 | 86.7% | 34% |
| Accompanied to the facility | 1748 | 21.9% | 41.4% |
| Transportation mode | | | |
| Walk | 1746 | 45.1% | 49.8% |
| Bicycle | 1746 | 6.9% | 25.4% |
| Your own motorbike | 1746 | 4.3% | 20.3% |
| Motorbike taxi | 1746 | 29.8% | 45.7% |
| Your own car | 1746 | 2.7% | 16.4% |
| Bus | 1746 | 8.4% | 27.8% |
| Bajaj | 1746 | 2% | 14% |
| A friend or family member brought me | 1746 | 0.3% | 5.9% |
| Other | 1746 | 0.4% | 6.3% |
| Travel time (return) (min) | 1744 | 98.64 | 91.31 |
| Clinic time (min) | 1714 | 130.94 | 104.17 |
| Travel cost (US$D) | 1675 | 1.39 | 2.81 |

1: *n* of participants' last time seeking chronic disease (CD) care differs from *n* of patients with CDs because for 30 participants, the day of the survey was the day of their first diagnosis.
TB, tuberculosis.

due to their illness during the month before the survey, with almost 10% completely unable to work.

Most patients with CDs appeared adherent to their treatment and follow-up schedule, with 75% seeking care within the last month and 89% received prescription medicines during their last visit whereas 68% took their medicines within the last 7 days. Most participants were recruited from a health centre (67.1%) and 21.9% were accompanied to the facility, mostly by one of their children or their partner. Most participants travelled by foot (45%) or motorbike taxi (30%) for more than one and a half hours for the return journey. Including those who travelled by bicycle (6.9%), 52% of patients had no financial costs for travel. In addition to travel time, patients spent more than 2 hours at the facility, including both waiting and consultation time (table 2). The average travel time was longer when attending health centres or hospitals compared with dispensaries.

Total non-medical costs averaged US$2.26 (SD: US$3.02), with an average direct cost of US$1.39 (SD: US$2.81) and indirect cost of US$0.87 (SD: 0.56) (table 2). The subanalysis of self-reported forgone wages indicated an average opportunity cost of US$21.88 (SD: US$1180.83) (online supplemental table S2). Patients with NCD incurred significantly higher direct, indirect and total costs relative to other patients (table 2, figure 1). Furthermore, patients with comorbid chronic conditions spend more time seeking care compared with those with one condition and in turn bear higher opportunity costs. We also observed that those with insurance spend both more time travelling to the facility and more time at the facility. Sensitivity analyses indicated that with

**Table 2** Non-medical costs of care-seeking and productivity costs of illness (Direct non-medical costs are defined as those costs that are not part of the medical bills from the facility, but are still incurred when accessing care including costs for transportation, food and accommodation. In our study, only transportation costs are part of this category. Indirect non-medical costs refer to the time losses associated with seeking out medical care, including travel time and time spent at the facility waiting and receiving care. Even though the latter are also productivity losses, we specifically refer here to productivity costs (of illness) as the lost wages that are a result of not being able to work due to (chronic) illness.), estimated using the human capital method

| | Direct non-medical cost | | | Indirect non-medical cost | | | Total non-medical cost | | | Productivity costs of illness (patients with CD) | | |
|---|---|---|---|---|---|---|---|---|---|---|---|---|
| | N | Mean (SD) | P value | N | Mean (SD) | P value | N | Mean (SD) | P value | N | Mean (SD) | P value |
| All | 1675 | 1.39 (2.81) | | 1711 | 0.87 (0.56) | | 1638 | 2.26 (3.02) | | 534 | 10.44 (19.41) | |
| Chronic condition | | | *** | | | *** | | | *** | | | |
| ▲ No | 1165 | 1.22 (2.68) | | 1189 | 0.80 (0.50) | | 1140 | 2.03 (2.82) | | | . | |
| ▲ Yes | 510 | 1.76 (3.05) | | 522 | 1.01 (0.66) | | 498 | 2.79 (3.36) | | | . | |
| Multimorbidity | | | *** | | | *** | | | *** | | | *** |
| ▲ No | 1568 | 1.35 (2.82) | | 1603 | 0.84 (0.53) | | 1534 | 2.20 (3.00) | | 423 | 9.53 (18.92) | |
| ▲ Yes | 107 | 1.92 (2.54) | | 108 | 1.20 (0.84) | | 104 | 3.16 (3.12) | | 111 | 13.92 (20.87) | |
| Insurance status | | | *** | | | *** | | | *** | | | ** |
| ▲ Uninsured | 1012 | 1.42 (3.08) | | 1029 | 0.78 (0.53) | | 984 | 2.22 (3.28) | | 238 | 12.94 (21.37) | |
| ▲ Insured | 662 | 1.34 (2.33) | | 681 | 0.99 (0.58) | | 653 | 2.33 (2.57) | | 296 | 8.43 (17.45) | |
| Sex | | | | | | | | | | | | |
| ▲ Female | 1074 | 1.43 (2.81) | | 1092 | 0.87 (0.58) | | 1052 | 2.32 (3.04) | | 363 | 10.17 (19.18) | |
| ▲ Male | 601 | 1.31 (2.80) | | 619 | 0.85 (0.52) | | 586 | 2.17 (2.97) | | 171 | 11.01 (19.93) | |
| Accompanier | | | *** | | | *** | | | *** | | | ** |
| ▲ No | 1320 | 1.18 (2.62) | | 1335 | 0.84 (0.55) | | 1290 | 2.03 (2.82) | | 405 | 8.80 (17.65) | |
| ▲ Yes | 355 | 2.15 (3.32) | | 376 | 0.95 (0.57) | | 348 | 3.12 (3.52) | | 129 | 15.59 (23.44) | |
| Facility level | | | *** | | | *** | | | *** | | | |
| ▲ Dispensary | 158 | 0.64 (1.50) | | 166 | 0.52 (0.44) | | 158 | 1.15 (1.70) | | 31 | 3.79 (10.93) | |
| ▲ Health centre | 1124 | 1.46 (3.01) | | 1150 | 0.86 (0.52) | | 1101 | 2.34 (3.16) | | 327 | 11.29 (20.10) | |
| ▲ Hospital | 393 | 1.47 (2.58) | | 395 | 1.02 (0.64) | | 379 | 2.50 (2.92) | | 176 | 10.03 (19.09) | |
| Occupation | | | | | | ** | | | *** | | | *** |
| ▲ Formal | 210 | 1.11 (2.29) | | 227 | 0.81 (0.49) | | 207 | 1.92 (2.50) | | 51 | 1.99 (5.20) | |
| ▲ Farmer | 906 | 1.60 (3.24) | | 913 | 0.85 (0.58) | | 884 | 2.47 (3.44) | | 314 | 10.48 (19.05) | |
| ▲ Self-employed | 288 | 1.20 (2.26) | | 288 | 0.87 (0.58) | | 280 | 2.10 (2.54) | | 64 | 8.20 (17.08) | |
| ▲ Other | 266 | 1.10 (1.96) | | 278 | 0.94 (0.53) | | 262 | 2.03 (2.20) | | 100 | 14.70 (23.32) | |

*P<0.05, **p<0.01, ***p<0.001.

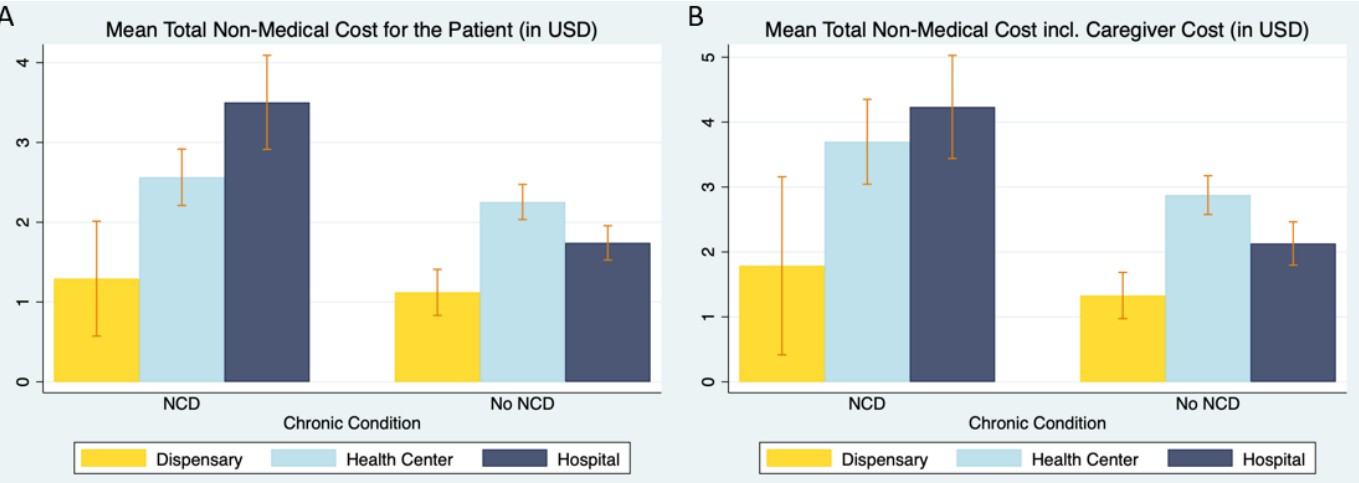

**Figure 1** (A) Total non-medical costs by health facility level and non-communicable disease (NCD) status. (B) Total non-medical costs by health facility level and NCD status, including caregiver cost.

the introduction of a 718 TZS minimum hourly wage, the increased value of time spent travelling and at the clinic leads to a 15.5% increase in the total non-medical cost of seeking care to US$2.91 per care-seeking episode (online supplemental table S2).

Our analysis using mixed-effect multivariate linear regression revealed that having health insurance, hypertension or multiple chronic conditions are associated with higher non-medical costs of care (table 3). Specifically, patients with health insurance pay on average 14% more than uninsured patients, while those with hypertension or multiple chronic conditions pay 14% and 35%

more, respectively. Higher education and seeking care at the closest facility to one's residence are associated with 11% lower non-medical costs. However, seeking care at a hospital (as opposed to a dispensary or health centre) is associated with a 67% increase in costs, and being accompanied to the facility is associated with a 39% increase in costs. In addition, we found a significant interaction effect for patients with hypertension who attend hospitals for their care, with costs being on average 35% higher than for other hospital outpatients (table 4).

We also observed that patients with multiple chronic conditions tend to miss more work days due to illness, while those with insurance or prescription medications tend to miss fewer. More highly educated individuals tend to miss fewer days, while older patients tend to miss more. However, we did not find any significant associations between treatment-seeking behaviour or medication adherence and absenteeism among patients with CDs.

**Table 3** Mixed-effect linear regression for total non-medical cost, including a random intercept for administrative district

| | Total non-medical cost | 95% CI |
|---|---|---|
| Health insurance | 0.137** (0.0504) | 0.04 to 0.24 |
| Hypertension | 0.143* (0.0717) | 0.002 to 0.28 |
| Multiple chronic conditions | 0.354*** (0.0998) | 0.16 to 0.55 |
| Sex (= male) | 0.00384 (0.0487) | 0.09 to 0.10 |
| Median age | −0.0870 (0.0535) | 0.19 to 0.02 |
| Higher education | −0.114** (0.0543) | 0.22 to 0.008 |
| Employed (last 12 months) | 0.0353 (0.0561) | 0.08 to 0.15 |
| Closest facility | −0.986*** (0.0584) | 1.10 to 0.87 |
| Accompanier | 0.390*** (0.0561) | 0.28 to 0.50 |
| Hospital | 0.673*** (0.0915) | 0.49 to 0.85 |
| Health centre | 0.567*** (0.0809) | 0.41 to 0.73 |
| Hospital#Hypertension | 0.352** (0.136) | 0.09 to 0.62 |
| Constant | 8.115*** (0.0993) | 7.92 to 8.31 |
| Observations | 1637 | |
| Clusters (districts) | 2 | |

SEs in parentheses.
95% CI: negative numbers in parentheses.
*P<0.05, **p<0.01, ***p<0.001.

## DISCUSSION
The results of this study reveal that patients still incur substantial non-medical costs when seeking healthcare, equivalent to 114% of the minimum daily wage (considering a minimum hourly wage of US$0.22 and 9 working hours per day results in a daily wage of US$1.98. The total non-medical cost was on average US$2.26; yielding a cost-to-wage ratio of approximately 114%). These findings are consistent with previous research conducted in Tanzania and in other SSA countries. One major contributor to these costs is the time required to reach the healthcare facilities. For instance, another study reported an average of 62 min to reach a hospital, compared with nearly 50 min in this study.[35] In a sample of 1407 patients requiring maternal and child healthcare (MCH), the average travel time was 30 min for a one-way trip, at an average cost of US$0.41.[36] In addition, the reported average time spent at the clinic was almost an hour. These estimates are somewhat lower than our finding, which could be

**Table 4** Negative binomial regression of the number of days in the past month that illness of patients with non-communicable disease prevented them from working

| | Number of days missed work | 95% CI |
|---|---|---|
| Health insurance | −0.539*** (0.139) | 0.81 to 0.27 |
| Multiple chronic conditions | 0.502*** (0.150) | 0.21 to 0.80 |
| Medicines prescribed | −0.782** (0.262) | 1.30 to 0.27 |
| Medicines taken (last 7 days) | 0.354 (0.201) | 0.04 to 0.75 |
| Last time sought care | | |
| ▶ Within the last 6 months | −0.423 (0.259) | 0.93 to 0.08 |
| ▶ More than 6 months ago | −0.448 (0.230) | 0.90 to 0.003 |
| Sex (=male) | 0.0380 (0.144) | 0.24 to 0.32 |
| Age | 0.0284*** (0.00573) | 0.02 to 0.04 |
| Higher education | −0.629** (0.207) | 1.03 to 0.22 |
| Occupation | | |
| ▶ Farmer | 0.461 (0.326) | 0.18 to 1.10 |
| ▶ Self-employed | 0.635 (0.363) | 0.08 to 1.35 |
| ▶ Other | 0.857* (0.338) | 0.19 to 1.52 |
| Constant | 0.190 (0.496) | 0.78 to 1.16 |
| Observations | 515 | |

SEs in parentheses.
95% CI: negative numbers in parentheses.
*P<0.05, **p<0.01, ***p<0.001.

due to the greater and more decentralised availability of MCH services that do not require patients to travel as far. In fact, 70% of those seeking MCH care walked to the facility,[36] which is far higher than those in our sample.

People living with HIV in rural Tanzania cross even larger distances, travelling an average of 2.81 hours in total and staying 2.32 hours at the clinic, but spend slightly less on transportation (US$1.09) than in our sample.[11] The opportunity cost of illness was also lower (US$3.79), with patients being ill for 16 hours per year on average, which is potentially due to the restriction of the sample to stable patients with HIV. Specifically for patients with cardiovascular diseases, another study reported annual transport costs of US$14 in rural areas and US$24 in urban areas, with average waiting times of 2 hours and 4 hours, respectively.[27] In addition, they reported annual income losses of US$23 and US$30 per year, respectively. However, these figures are difficult to compare to those of this study because we estimated costs per visit rather than annually. In addition, this study only had a rather small sample size of 100 patients, and included only four health facilities. Highly educated individuals incur lower time and travel costs in our sample, similar to other findings.[37] We suspect that this is a result of highly educated people being less likely to be engaged in strenuous physical labour and their proximity to health facilities, given that they tend to live in urban areas with high densities of facilities.

Our study revealed that health insurance status has contrasting effects on direct and indirect non-medical cost, with a non-significant negative association with travel cost and a strongly significant positive association with time cost, both with the travel and clinic time. This could result from the fact that a higher proportion of insured people seek care at the hospital. Since there are only a few hospitals, most people would have to travel further to reach them. In addition, they are often more crowded, leading to longer waiting times and higher time costs.[27]

The financial burden of seeking healthcare does not solely fall on patients, but also their informal caregivers. Our study shows that having an accompanier to bring patients to healthcare facilities is associated with significantly higher direct and indirect non-medical costs, even without accounting for the caregiver cost. This might be due to the most severely ill-being significantly more likely to require accompaniment in order to access the services they need. Those accompanied by an informal caregiver are often unable to walk to the facility and the care they need may not be available at their nearest facility, resulting in higher travel and time costs (table 2). The lack of previous studies with which to compare our findings demonstrates the novelty of our work. However, a study conducted in Ghana supports the interpretation of our findings in that patients with CDs were shown to use accompaniers in order to overcome mobility and transport barriers to reaching more distant tertiary care facilities,[38] thus explaining why those with accompaniers incur higher direct and indirect non-medical costs than those without. Additionally, the majority of participants reported their coresident household members, such as their partner or child, as informal caregivers, indicating that these households face a double burden.

Patients with CDs face such costs even more frequently than other patients, due to the need for monthly treatment and the frequent stockouts of essential medicines, which potentially explains why most survey participants' last doctor's visit occurred within the last month.[13 39] Therefore, improving accessibility of NCD care at primary and secondary care levels can reduce non-medical costs and improve the availability of medicines at these levels of care. This would allow patients to receive longer-term prescriptions, requiring fewer visits to the healthcare facility to refill medications. Decentralising NCD services would not only provide more affordable care for patients with NCD but would result in cost savings for insurers and the health system,[40] while generating additional revenue for primary and secondary care facilities.

Furthermore, it is noteworthy that the opportunity cost associated with illness-related missed work surpasses the non-medical cost for accessing care, and is significantly higher for uninsured patients than insured ones, as they on average miss 2 more days of work per month. Absenteeism thus affects the uninsured more than the insured, knowing that direct OOP payments introduce a substantial financial barrier to accessing care.[41] However, our results suggest that engagement in care is associated with

fewer missed work days. Moreover, retired, and hence older individuals, are most often unable to carry out their daily tasks, while those in formal employment seem to be the least impacted.

Our estimates of absenteeism due to chronic illness are higher than those reported in other studies. For instance, a study in Namibia found an average of only 1 day of sick leave over a 90-day period in employer records.[42] However, their study included both sick and non-sick employees, while our absenteeism data focused only on patients with chronic illnesses. Additionally, employer records may not capture the missed workdays of informal workers who make up the majority of Tanzania's workforce, particularly in the agriculture sector, where physically strenuous work is more likely to be impacted by chronic illness. To reduce the access and opportunity costs for patients with NCD, it would be important to prioritise the provision of basic NCD services at health centres and dispensaries and actively promote patients' engagement in care.

The study findings should be interpreted in the light of some limitations. The key limitation is the lack of reliable self-reported individual income estimates, which required us to rely on the HCM. In the online supplemental materials, we provided an overview of the costs, as presented in table 2, including the self-reported lost income due to seeking care on the day of the survey. These figures suggest that our estimates potentially underestimate the indirect non-medical cost and opportunity costs. However, participants are likely to overestimate their lost income, so we suspect that the actual cost lies somewhere in between. Other limitations are that travel, clinic time and number of missed days of work were self-reported and that the use of multiple modes of transportation was not considered when assessing travel costs. Additionally, exit surveys do not capture individuals with NCDs who do not seek healthcare at all, potentially hindered by the high cost burden. However, it does minimise recall bias related to the reported time and cost variables. While minimum wages have increased since the data collection for this study in 2020/2021, corresponding acceleration of inflation and fuel price increases may have negated any potential improvements to the affordability of the direct non-medical costs of seeking care.

In addition, we were unable to explore distress-financing strategies such as borrowing money or selling assets, nor could we stratify costs by socioeconomic status. Lastly, research has shown that the direct and indirect costs of seeking care are higher during the rainy season[43]; as our data collection occurred during the dry season, we would expect the travel time to be higher during other times of the year.[44] Given that most patients were subsistence farmers, they potentially not only experience seasonality in their income, but also in their medical costs, which was not captured in this study.

Despite these limitations, this study sheds light on the challenges faced by Tanzanian patients with chronic conditions seeking care at health facilities, especially in rural districts. The study suggests that decentralising the provision of NCD care from hospitals to health centres and dispensaries may be beneficial. This approach could help to reduce patients' non-medical and opportunity costs associated with travel and increase their engagement in care.

**Acknowledgements** The authors wish to acknowledge the field workers of the Ifakara Health Institute for their data collection efforts, and the stakeholders and policy-makers who attended dissemination workshops in Dodoma and Dar es Salaam whose discussions provided valuable input and context to this work.

**Contributors** AV, BH and FT developed the research design. AV and BH drafted the initial version of the manuscript and performed the analysis under the supervision of FT. KT and GM provided inputs on the methodology and implemented the data collection. All authors contributed to the article and approved the submitted version. AV and BH are joint first authors of this work. FT is responsible for the overall content as the guarantor.

**Funding** This work was funded by the Swiss Programme for Research on Global Issues for Development (Grant number: 183760), a joint funding initiative by the Swiss Agency for Development and Cooperation (SDC) and the Swiss National Science Foundation (SNSF). The funders had no role in study design, data collection and analysis, decision to publish, or preparation of the manuscript.

**Competing interests** None declared.

**Patient and public involvement** Patients and/or the public were involved in the design, or conduct, or reporting, or dissemination plans of this research. Refer to the Methods section for further details.

**Patient consent for publication** Not applicable.

**Ethics approval** This studied obtained ethical clearance from the Ifakara Health Institute (IHI) Institutional Review Board (IHI/IRB/EXT/No:35–2020) and the Tanzanian National Institute for Medical Research (NIMR/HQ/R.8a/Vol.IX/3518). Participants gave informed consent to participate in the study before taking part.

**Provenance and peer review** Not commissioned; externally peer reviewed.

**Data availability statement** Data are available upon reasonable request. Data are available upon reasonable request. According to the institutional review board of IHI, we are not allowed to make the data publicly available. Interested researchers should contact the corresponding author.

**ORCID iDs**
Brady Hooley http://orcid.org/0000-0003-1948-9654
Grace Mhalu http://orcid.org/0000-0002-0417-3027
Fabrizio Tediosi http://orcid.org/0000-0001-8671-9400

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
