## [Reviewer comments · BMJ Open]

ARTICLE DETAILS

TITLE (PROVISIONAL)	A cross-sectional study of the burden and determinants of non-medical and opportunity costs of accessing chronic disease care in rural Tanzania
AUTHORS	Verjans, Anna; Hooley, Brady; Tani, Kassimu; Mhalu, Grace; Tediosi, Fabrizio

VERSION 1 – REVIEW

REVIEWER	Shukla , Varsha Birla Institute of Technology and Science - Pilani Campus
REVIEW RETURNED	27-Oct-2023

GENERAL COMMENTS	1. The last line of the objective “explored the relation between engagement in care and NCD related work limitation” is not in line with title, research question and results. Therefore, should be revised. The objectives should be revised with more clarity.2. Similarly, the answers to the second and third question in the key message should be revised in line with the conclusion. Non-communicable disease is the concerned problem whereas health insurance is a financial solution therefore cannot be used in the same context.3. The objective mentions about the comparison with non-NCD patient, however the same is not briefed in the results.4. The methodology should be elaborated more.5. Please define all abbreviations at first mention in the text. Page no. 5, ART.6. The study should list the NCDs it focused for the analysis.7. References 1, 24-28 should be revised with proper format and URL address.8. The manuscript needs to be edited to improve the English/grammar throughout.9. The plagiarism percentage should be confirmed by the editors.10. Could there be a comparison with the existing literature and discussion on the novelty of the study in the field.11. Overall, it is a good attempt in the era of rising NCDs and related financial burden around the globe. The author can improve by research article by clearly stating the objectives, and organizing the structure paper in line with the research problem in hand.
---

REVIEWER	Abraham, Sunil Christian Medical College and Hospital Vellore, Family Medicine
REVIEW RETURNED	30-Oct-2023

GENERAL COMMENTS	217 Was the 5 working days lost in addition to the days lost for travel to the health centre? In other words, were the 5 days lost due to an effect of the disease or travel for treatment?
---

	Only 26.8% of the total patients who were surveyed were working. Is there data of the percentage of patients with NCDs who were working? Since the study is looking at loss of income due to NCDs, not clear why students are listed under working category. Same goes for retired patients and those taking care of home and children. There should be no change in the income of these categories of patients due to the visit to the health centre 220 Interesting to note that most NCD patients appeared to adhere to treatment and follow up even in the face of additional expenses involved. Shows that patients were committed to the continuity of treatment. Table 2: Need to define what is Direct Non-Medical Cost, Indirect Non-Medical Cost, Total Non-Medical Cost and Estimated Lost Income 234 Need to define what is Direct Non-Medical Cost, Indirect Non-Medical Cost, Total Non-Medical Cost and Estimated Lost Income 237 Have you done an analysis of the location of services used by those with insurance and do not have insurance? 244 It is intriguing that having health insurance might increase money spent as non-medical costs of care. Is it because more people access health care facilities now that the health insurance is covering part of that expense? 80.7% of the patients had a diagnosis of hypertension. How does the change in diagnosis change the non medical expenses? Is it because they had to make more trips? Same question as above- why do patients with multiple chronic conditions have to pay more- is it because there was a change in transportation as they were too ill and needed more expensive methods of transport? Have you looked at the mode of transportation involved in non medical costs? For example a patient with more severe disease might need a car while others might travel by bike and reduce the cost of transportation. 303. The term " lower levels of care" makes it sound as if the care is of lower quality or lower priority. From the patient's perspective, care that is accessible closer to home will be the highest level of care.." Community based care" might be a better terminology to describe a level of care that is based in the community and accessible to the population. 308 Not clear if it is the insured or the uninsured patients who end up spending more. This seems to contradict the earlier statement that those with insurance spend more. 310 "OOP". Need to expand to "out of pocket expenses" as it is mentioned for the first time 217 Was the 5 working days lost in addition to the days lost for travel
--	---

	to the health centre? In other words, were the 5 days lost due to an effect of the disease or travel for treatment? Only 26.8% of the total patients who were surveyed were working. Is there data of the percentage of patients with NCDs who were working? Since the study is looking at loss of income due to NCDs, not clear why students are listed under working category. Same goes for retired patients and those taking care of home and children. There should be no change in the income of these categories of patients due to the visit to the health centre 220 Interesting to note that most NCD patients appeared to adhere to treatment and follow up even in the face of additional expenses involved. Shows that patients were committed to the continuity of treatment. Table 2: Need to define what is Direct Non-Medical Cost, Indirect Non-Medical Cost, Total Non-Medical Cost and Estimated Lost Income 234 Need to define what is Direct Non-Medical Cost, Indirect Non-Medical Cost, Total Non-Medical Cost and Estimated Lost Income 237 Have you done an analysis of the location of services used by those with insurance and do not have insurance? 244 It is intriguing that having health insurance might increase money spent as non-medical costs of care. Is it because more people access health care facilities now that the health insurance is covering part of that expense? 80.7% of the patients had a diagnosis of hypertension. How does the change in diagnosis change the non medical expenses? Is it because they had to make more trips? Same question as above- why do patients with multiple chronic conditions have to pay more- is it because there was a change in transportation as they were too ill and needed more expensive methods of transport? Have you looked at the mode of transportation involved in non medical costs? For example a patient with more severe disease might need a car while others might travel by bike and reduce the cost of transportation. 303. The term " lower levels of care" makes it sound as if the care is of lower quality or lower priority. From the patient's perspective, care that is accessible closer to home will be the highest level of care.." Community based care" might be a better terminology to describe a level of care that is based in the community and accessible to the population. 308 Not clear if it is the insured or the uninsured patients who end up spending more. This seems to contradict the earlier statement that those with insurance spend more. 310 "OOP". Need to expand to "out of pocket expenses" as it is mentioned for the first time
--	---

	STROBE statement : Not explained how the study size was arrived at The statistical method has to be reviewed by a specialist
--	--

REVIEWER	Odunyemi, Adedokun Murdoch University College of Arts Business Law and Social Sciences
REVIEW RETURNED	07-Nov-2023

GENERAL COMMENTS	Comments This study used a cross-sectional study to detail the substantial burden placed on patients with chronic diseases, especially NCDs, by medical and non-medical indirect costs of seeking care in Tanzania. It compares patients with chronic diseases with other patients, evaluates the factors that propel the differences, and explores the relationship between NCD care and productivity loss. The findings indicate that NCD-related care imposed higher indirect costs of care. Multimorbidities, old age, being accompanied by caregivers, and hypertension, especially hospital care, were associated with increased indirect costs of care and lost workdays. However, health insurance lowered the number of lost workdays, increased indirect costs, and lowered higher education. Major Comments: 1. Introduction The introduction is well-written, providing an argument that sets the context for the study using relevant literature and highlighting existing gaps and the importance and contribution of the study. Though some of the supporting evidence is older than ten years, the majority are current. The authors might replace those old citations with newer ones to improve the originality and topicality of the argument. 2. Methods The methods used are relevant and well-explained, including the theories and statistical methods. The rationale for the choice of statistical method is clearly stated. Using control and sensitivity analysis and accounting for variability increase the validity and robustness of the findings. However, there is no data availability statement, which impugns the transparency of the study. Moreover, this section should briefly highlight any bias associated in the data and methodology used. 3. Results (a) Text: The result of the study is well-presented and flows logically. However, some areas require improvement in the presentation of the result. They are as follows:  • Since approximately 12% of the chronic conditions in your sample are infectious diseases and others, it is pertinent to explain that your analysis was on chronic diseases, of which the largest percentage is NCDs. • The statistics about patients' adherence to medication and treatment on page 9, lines 220-222, are not referenced in any table or graph. The tables and graphs must be a standalone from the text. • I suggest the statement "Most participants travelled by foot (45%) or motorbike taxi (30%)" should be improved to differentiate two categories of participants: those that travelled by foot and bicycle, constituting 52% (which did not incur transportation cost, but more travelling time) and those using motorbike taxi, comprising the
---

	majority (30%) of those incurring transportation costs and lesser travelling time. The importance of these distinctions is that while the former group contributed to non-medical indirect costs, the latter contributed significantly to medical indirect costs. The policy relevance of both groups is contained in the provision of a good road network and affordable commercial transport system.  • The reason statement “which could result from the fact that a higher proportion of insured people seek care at the hospital and higher facility levels have significantly higher access costs” on page 12, lines 236-238, is irrelevant here. It should be moved to the discussion section. • Although hypertension contributed to the increasing indirect costs of care, the result shown in Table 3 reveals that, except for hospital care, this association was not statistically significant. This should be indicated in the statement on page 12, lines 241-243. • The outcome should be a critical analysis of the data collected (b) Data Table:  • Please supply the missing article “the” before “total” in Table 1’s title • It is unclear what item in Table 1 is referred to as “Normal facility visited”. • The “from” in Table 4’s title should come after “them 4. Discussion: The summary, contributions, significance, and limitations of the study and its findings are succinctly articulated in the Discussion. However, the following points should be noted:  • The elements being referred to in the first part of the sentence, In addition, patients reported an annual loss of income of \$23 and \$30 per year, respectively,” on Page 15, Lines 279-280, are not clearly stated. • The probable reason given for why patients with insurance incurred more indirect costs on Page 15, Lines 285-288 appears contradictory. The reason given showed that more patients with health insurance reside in urban areas where they have easier access to healthcare facilities, thus lowering their indirect travel costs. • It would be great if the finding highlighted on Page 15, Lines 291-293, could be compared to any extant studies. 5. Conclusions: The conclusion flows from the limitations of the study. It summarizes relevant findings and provides plausible contributions to policy. However, I suggest the statement “this study sheds light on the challenges faced by Tanzanian NCD patients seeking care at health facilities, particularly for those without insurance” on Page 15, Lines 345, be reworded since no finding particularly mentions the uninsured. Minor Comments: The manuscript is written in clear and concise language with very few grammatical and syntax errors.  • However, punctuation marks are consistently misplaced before instead of after in-text citations. • Please correct the subject-verb mismatch on page 15, line 293. “Data” is a plural word. • The expression “more than 20% of NCD patients” on page 9, line 215, would be more precise using “approximately 21% of NCD patients” or similar expressions. • The correct preposition after “accessibility” on Page 15, Line 302 is
--	--

VERSION 1 – AUTHOR RESPONSE

Reviewer: 1

Dr. Varsha Shukla , Birla Institute of Technology and Science - Pilani Campus

Comments to the Author:

1. The last line of the objective “explored the relation between engagement in care and NCD related work limitation” is not in line with title, research question and results. Therefore, should be revised. The objectives should be revised with more clarity.

Authors' response:

Many thanks for spotting this inaccuracy. We have revised the objectives accordingly.

Similarly, the answers to the second and third question in the key message should be revised in line with the conclusion. Non-communicable disease is the concerned problem whereas health insurance is a financial solution therefore cannot be used in the same context.

Authors' response:

Thank you for your comment. The editor asked us to remove the 'Key Messages' section and to replace it with a 'Strengths and limitations' section, specifically pertaining to the methods.

2. The objective mentions about the comparison with non-NCD patient, however the same is not briefed in the results.

Authors' response:

Thank you for your comment. We have addressed this in line 55: Patients with chronic conditions incurred significantly higher non-medical costs than other patients, with an average of \$2.79 (3.36) compared to \$2.03 (2.82).

3. The methodology should be elaborated more.

Authors' response:

To address this comment we have amended the methods section in several parts.

4. Please define all abbreviations at first mention in the text. Page no. 5, ART.

Authors' response:

Done

5. The study should list the NCDs it focused for the analysis.

Authors' response:

Thank you for your comment. The NCDs that were included in the analysis are listed in Table 1.

6. References 1, 24-28 should be revised with proper format and URL address.

Authors' response:

Thank you for your comment. We have revised these references.

7. The manuscript needs to be edited to improve the English/grammar throughout.

Authors' response:

The manuscript has been proof edited by a native English speaker.

8. The plagiarism percentage should be confirmed by the editors.

Authors' response:

Thank you for your suggestion. This manuscript is an original research study.

9. Could there be a comparison with the existing literature and discussion on the novelty of the study in the field.

Authors' response:

To address this comment we have amended the discussion section highlighting in more details how the results of this study compare with those of other similar studies.

10. Overall, it is a good attempt in the era of rising NCDs and related financial burden around the globe. The author can improve by research article by clearly stating the objectives, and organizing the structure paper in line with the research problem in hand.

Authors' response:

Many thanks for appreciating the relevance of this manuscript. We have amended the manuscripts to address all comments of reviewers and we think your suggestions have improved it substantially.

Reviewer: 2

Prof. Sunil Abraham, Christian Medical College and Hospital Vellore

Comments to the Author:

1. 217 Was the 5 working days lost in addition to the days lost for travel to the health centre? In other words, were the 5 days lost due to an effect of the disease or travel for treatment?

Authors' response:

The 5 days lost were due to the disease and therefore additional time lost to the travel time (lines 173-175 of the original submission).

2. Only 26.8% of the total patients who were surveyed were working. Is there data of the percentage of patients with NCDs who were working?

Since the study is looking at loss of income due to NCDs, not clear why students are listed under working category.

Same goes for retired patients and those taking care of home and children. There should be no change in the income of these categories of patients due to the visit to the health centre

Authors' response:

Among chronic disease patients 22.10% were working. Even for people whose primary occupation is being a student or those who are retired, they may also need to continue working to 'make ends meet'. Even if students or retired people are not working and they therefore don't lose income, their time still has intrinsic value and represents an opportunity cost, which is why we used the human capital method to quantify the value of lost time. Likewise, those taking care of the home and children are performing unpaid work that still has value and is accounted for under the human capital method of measuring the value of lost time. Accordingly, we have rephrased this from 'lost income' to 'opportunity costs'.

3. 220 Interesting to note that most NCD patients appeared to adhere to treatment and follow up even in the face of additional expenses involved. Shows that patients were committed to the continuity of treatment.

Author's response: We think this finding was expected in the sense that we performed exit surveys and therefore only capture those patients who go to the healthcare facility. Since those not seeking care were by definition not included, there are probably many chronic patients who do not continue treatment, but we would need household surveys to investigate this.

4. Table 2: Need to define what is Direct Non-Medical Cost, Indirect Non-Medical Cost, Total Non-Medical Cost and Estimated Lost Income

Authors' response:

To address this comment we have added a footnote to the table – with the definitions.

5.

234 Need to define what is Direct Non-Medical Cost, Indirect Non-Medical Cost, Total Non-Medical Cost and Estimated Lost Income

Authors' response: See footnote under table 2.

We have added the definitions

6.

237 Have you done an analysis of the location of services used by those with insurance and do not have insurance?

Authors' response: If you mean by 'location' the type of facility then we observed in our analysis that among the insured more patients seek care at the hospital compared to the uninsured. Since hospitals are usually located in urban areas, we see that among the insured more patients seek care at urban facilities compared to uninsured.

7. 244 It is intriguing that having health insurance might increase money spent as non-medical costs of care. Is it because more people access health care facilities now that the health insurance is covering part of that expense?

Authors' response:

Actually no, the cost we are looking at is per person and relates to the cost incurred when accessing the facility on the day of the survey (we do not ask about costs for previous visits). We see that the cost is higher for insured people because they seem to be spending more time at the facility. We assumed (and this is also confirmed by the data) that this is because insured people are more likely to seek care at hospitals rather than health centers or dispensaries, with hospitals tending to have longer waiting and travel times.

8. 80.7% of the patients had a diagnosis of hypertension. How does the change in diagnosis change the non medical expenses? Is it because they had to make more trips?

Authors' response: Actually no, we only have the costs for one visit and we are thus only accounting for the costs that patients incurred for one trip to the health facility. Hence, the cost is higher for patients with hypertension each time they access a health facility. It is indeed as you suggest below, we see that chronic patients tend to be older, need accompaniment to the facility and thus need more expensive modes of transportation.

9. Same question as above- why do patients with multiple chronic conditions have to pay more- is it because there was a change in transportation as they were too ill and needed more expensive methods of transport?

Authors' response: See response to the previous comment.

10. Have you looked at the mode of transportation involved in non medical costs? For example a patient with more severe disease might need a car while others might travel by bike and reduce the cost of transportation.

Authors' response: Yes, we did, indeed as you suggest we see that chronic patients tend to incur higher transportation costs due to more expensive modes of transport. As we expect, they tend to be older and more severely ill.

11. 303. The term " lower levels of care" makes it sound as if the care is of lower quality or lower priority. From the patient's perspective, care that is accessible closer to home will be the highest level of care.." Community based care" might be a better terminology to describe a level of care that is based in the community and accessible to the population.

Authors' response:

Thanks for this suggestion. We have amended the text replacing "lower levels of care" with dispensaries and health centres.

12.

308 Not clear if it is the insured or the uninsured patients who end up spending more. This seems to contradict the earlier statement that those with insurance spend more.

Authors' response:

Apologies if this was not clear. Insured patients spend more in total, but if we look at the direct and indirect non-medical costs separately, we see that they spend marginally less money on transportation (not statistically significant), but they have higher time costs as they spend more time at the facility. However, since the higher time cost surpasses the lower transportation cost, overall the insured spend more. We amended this paragraph and hope that it is clearer now.

13. 310 "OOP". Need to expand to "out of pocket expenses" as it is mentioned for the first time

Authors' response:

Thank you for your comment. We actually mentioned it in full in the original text on line 98, but we now also added the abbreviation.

14. STROBE statement :

Not explained how the study size was arrived at

Authors' response:

We added this, thank you, see footnote of page 6.

The statistical method has to be reviewed by a specialist

Authors' response:

Thank you for your suggestion. We have revised the methods and the section has been reviewed by a biostatistician that is now acknowledges in the acknowledgements

Reviewer: 3

Dr. Adelakun Odunyemi, Murdoch University College of Arts Business Law and Social Sciences
Please also see attached document.

Comments to the Author:

Comments

This study used a cross-sectional study to detail the substantial burden placed on patients with chronic diseases, especially NCDs, by medical and non-medical indirect costs of seeking care in Tanzania. It compares patients with chronic diseases with other patients, evaluates the factors that propel the differences, and explores the relationship between NCD care and productivity loss. The

findings indicate that NCD-related care imposed higher indirect costs of care. Multimorbidities, old age, being accompanied by caregivers, and hypertension, especially hospital care, were associated with increased indirect costs of care and lost workdays. However, health insurance lowered the number of lost workdays, increased indirect costs, and lowered higher education.

Major Comments:

1. Introduction

The introduction is well-written, providing an argument that sets the context for the study using relevant literature and highlighting existing gaps and the importance and contribution of the study. Though some of the supporting evidence is older than ten years, the majority are current. The authors might replace those old citations with newer ones to improve the originality and topicality of the argument.

Authors' response:

Many thanks for this suggestion. We reviewed the references and added a few recent ones that we had missed in the submitted version.

2. Methods

The methods used are relevant and well-explained, including the theories and statistical methods. The rationale for the choice of statistical method is clearly stated. Using control and sensitivity analysis and accounting for variability increase the validity and robustness of the findings. However, there is no data availability statement, which impugns the transparency of the study. Moreover, this section should briefly highlight any bias associated in the data and methodology used.

Authors' response:

Many thanks for this suggestion. To address this comment we have added a data availability statement and amended the methods section to highlight the sampling bias.

3. Results

(a) Text:

The result of the study is well-presented and flows logically. However, some areas require improvement in the presentation of the result. They are as follows:

- Since approximately 12% of the chronic conditions in your sample are infectious diseases and others, it is pertinent to explain that your analysis was on chronic diseases, of which the largest percentage is NCDs.

Authors' response:

Many thanks for this suggestion that we have taken up amending the text accordingly. We have now changed NCDs to chronic diseases, since this is indeed more correct.

- The statistics about patients' adherence to medication and treatment on page 9, lines 220-222, are not referenced in any table or graph. The tables and graphs must be a standalone from the text.

Authors' response:

Many thanks for this suggestion. We have amended the tables, such that they are standalone.

- I suggest the statement "Most participants travelled by foot (45%) or motorbike taxi (30%)" should be improved to differentiate two categories of participants: those that travelled by foot and bicycle, constituting 52% (which did not incur transportation cost, but more travelling time) and those using motorbike taxi, comprising the majority (30%) of those incurring transportation costs and lesser travelling time. The importance of these distinctions is that while the former group contributed to non-

medical indirect costs, the latter contributed significantly to medical indirect costs. The policy relevance of both groups is contained in the provision of a good road network and affordable commercial transport system.

Authors' response:

Many thanks for this suggestion. We actually do not observe shorter travel times for those travelling by motorbike taxi (or another form of paid transportation). Those who are taking paid transportation potentially live further away from the facility and therefore do not necessarily have shorter travel times, while those who walk or bike might live closer to their nearest facility.

- The reason statement "which could result from the fact that a higher proportion of insured people seek care at the hospital and higher facility levels have significantly higher access costs" on page 12, lines 236-238, is irrelevant here. It should be moved to the discussion section.

Authors' response:

Many thanks for this suggestion. We have moved the sentence to the discussion section.

- Although hypertension contributed to the increasing indirect costs of care, the result shown in Table 3 reveals that, except for hospital care, this association was not statistically significant. This should be indicated in the statement on page 12, lines 241-243.

Authors' response:

Thank you for your suggestion. However, the regression coefficients in Table 3 are to be interpreted keeping all other factors constant. Therefore, in Table 3 we can observe that hypertension has both a significant main effect at an $\alpha=0.05$ level and a significant interaction effect with hospital-based care, when keeping all other variables constant (note that one asterisk represents significance at the $\alpha=0.05$ level in our table). Including interaction effects is important to understand whether a variable's effect is significant on its own, or only when interacted with another variable. Here we thus show that even though the interaction effect is significant, both hypertension and health facility level have statistically significant main associations with non-medical costs on their own.

- The outcome should be a critical analysis of the data collected

(b) Data Table:

- Please supply the missing article "the" before "total" in Table 1's title
- It is unclear what item in Table 1 is referred to as "Normal facility visited".
- The "from" in Table 4's title should come after "them"

Authors' response:

Many thanks for these comments. We have made the changes accordingly.

4. Discussion:

The summary, contributions, significance, and limitations of the study and its findings are succinctly articulated in the Discussion. However, the following points should be noted:

- The elements being referred to in the first part of the sentence, "In addition, patients reported an annual loss of income of \$23 and \$30 per year, respectively," on Page 15, Lines 279-280, are not clearly stated.
- The probable reason given for why patients with insurance incurred more indirect costs on Page 15, Lines 285-288 appears contradictory. The reason given showed that more patients with health insurance reside in urban areas where they have easier access to healthcare facilities, thus lowering their indirect travel costs.
- It would be great if the finding highlighted on Page 15, Lines 291-293, could be compared to any

extant studies.

Authors' response:

Many thanks for these remarks.

- We have adjusted the sentence and we hope it is clearer now.
- Thank you for pointing this out. We have adapted this paragraph now to improve consistency.
- Unfortunately, we could not find any studies to compare our finding to on the direct and indirect non-medical costs for patients with caregivers. However, we made sure to highlight this lack of research and referred to one study that looks into the use of accompaniers among adults with chronic diseases to reach more distant facilities. (see line 316-319)

5. Conclusions:

The conclusion flows from the limitations of the study. It summarizes relevant findings and provides plausible contributions to policy. However, I suggest the statement “this study sheds light on the challenges faced by Tanzanian NCD patients seeking care at health facilities, particularly for those without insurance” on Page 15, Lines 345, be reworded since no finding particularly mentions the uninsured.

Authors' response:

Many thanks for this suggestion. We have left out the part about “those without insurance”.

Minor Comments:

The manuscript is written in clear and concise language with very few grammatical and syntax errors.

- However, punctuation marks are consistently misplaced before instead of after in-text citations.
- Please correct the subject-verb mismatch on page 15, line 293. “Data” is a plural word.
- The expression “more than 20% of NCD patients” on page 9, line 215, would be more precise using “approximately 21% of NCD patients” or similar expressions.
- The correct preposition after “accessibility” on Page 15, Line 302 is “to” and not “of”.

Authors' response: Thank you for these comments. Only the first point on the in-text citations we could unfortunately not change, since this was a requirement of the journal and is part of the Vancouver citation style. With regards to accessibility, it would be correct to refer to “access to” something, however “accessibility of” remains correct in our case.

VERSION 2 – REVIEW

REVIEWER	Odunyemi, Adalakun Murdoch University College of Arts Business Law and Social Sciences
REVIEW RETURNED	17-Jan-2024
GENERAL COMMENTS	The authors have adequately addressed all of my comments, and I am satisfied with their responses. Therefore, I recommend that the manuscript is suitable for publication.